# 3D Former: Monocular Scene Reconstruction with 3D SDF Transformers

**Weihao Yuan, Xiaodong Gu, Heng Li, Zilong Dong, Siyu Zhu**[*]
Alibaba Group
`{qianmu.ywh, dadong.gxd, baoshu.lh, list.dzl, siting.zsy}`
`@alibaba-inc.com`

## Abstract

Monocular scene reconstruction from posed images is challenging due to the complexity of a large environment. Recent volumetric methods learn to directly predict the TSDF volume and have demonstrated promising results in this task. However, most methods focus on how to extract and fuse the 2D features to a 3D feature volume, but none of them improve the way how the 3D volume is aggregated. In this work, we propose an SDF transformer network, which replaces the role of 3D CNN for better 3D feature aggregation. To reduce the explosive computation complexity of the 3D multi-head attention, we propose a sparse window attention module, where the attention is only calculated between the non-empty voxels within a local window. Then a top-down-bottom-up 3D attention network is built for 3D feature aggregation, where a dilate-attention structure is proposed to prevent geometry degeneration, and two global modules are employed to equip with global receptive fields. The experiments on multiple datasets show that this 3D transformer network generates a more accurate and complete reconstruction, which outperforms previous methods by a large margin. Remarkably, the mesh accuracy is improved by $41.8\%$, and the mesh completeness is improved by $25.3\%$ on the ScanNet dataset. [1]

## 1 Introduction

Monocular 3D reconstruction is a classical task in computer vision and is essential for numerous applications like autonomous navigation, robotics, and augmented/virtual reality. Such a vision task aims to reconstruct an accurate and complete dense 3D shape of an unstructured scene from only a sequence of monocular RGB images. While the camera poses can be estimated accurately with the state-of-the-art SLAM (Campos et al., 2021) or SfM systems (Schonberger & Frahm, 2016), a dense 3D scene reconstruction from these posed images is still a challenging problem due to the complex geometry of a large-scale environment, such as the various objects, flexible lighting, reflective surfaces, and diverse cameras of different focus, distortion, and sensor noise. Many previous methods reconstruct the scenario in a multi-view depth manner (Yao et al., 2018; Chen et al., 2019; Duzceker et al., 2021). They predict the dense depth map of each target frame, which can estimate accurate local geometry but need additional efforts in fusing these depth maps (Murez et al., 2020; Sun et al., 2021), e.g., solving the inconsistencies between different views.

Recently, some methods have tried to directly regress the complete 3D surface of the entire scene (Murez et al., 2020; Sun et al., 2021) from a truncated signed distance function (TSDF) representation. They first extract the 2D features with 2D convolutional neural networks (CNN), and then back-project the features to 3D space. Afterward, the 3D feature volume is processed by a 3D CNN network to output a TSDF volume prediction, which is extracted to a surface mesh by marching cubes (Lorensen & Cline, 1987). This way of reconstruction is end-to-end trainable, and is demonstrated to output accurate, coherent, and complete meshes. In this paper, we follow this volume-based 3D reconstruction path and directly regress the TSDF volume.

---

[*]Corresponding Author
[1]Project Page: `https://weihaosky.github.io/former3d`

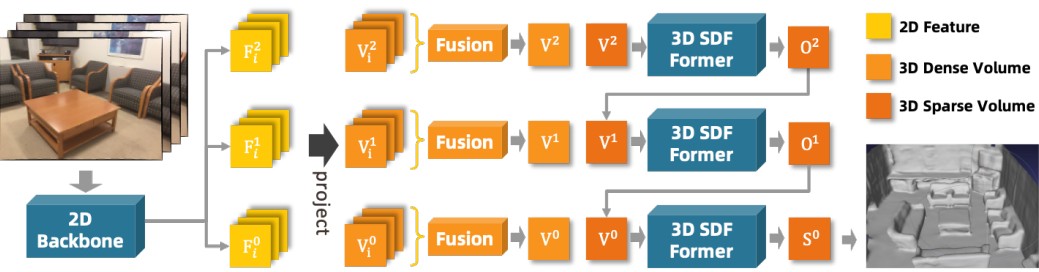

Figure 1: The overview of the 3D reconstruction framework. The input images are extracted to features by a 2D backbone network, then the 2D features are back-projected and fused to 3D feature volumes, which are aggregated by our 3D SDF transformer and generate the reconstruction in a coarse-to-fine manner.

Inspired by recent successes of vision transformer (Vaswani et al., 2017; Dosovitskiy et al., 2020), some approaches (Bozic et al., 2021; Stier et al., 2021) have adopted this structure in 3D reconstruction, but their usages are all limited to fusing the 2D features from different views while the aggregation of the 3D feature volumes is still performed by the 3D CNN. In this paper, we claim that the aggregation of 3D feature volume is also critical, and the evolution from 3D CNN to 3D multi-head attention could further improve both the accuracy and completeness of the reconstruction. Obviously, the limited usage of 3D multi-head attention in 3D feature volume aggregation is mainly due to its explosive computation. Specifically, the attention between each voxel and any other voxel needs to be calculated, which is hard to be realized in a general computing platform. This is also the reason why there are only a few applications of 3D transformers in solving 3D tasks.

In this work, to address the above challenges and make the 3D transformer practical for 3D scene reconstruction, we propose a sparse window multi-head attention structure. Inspired by the sparse CNN (Yan et al., 2018), we first sparsify the 3D feature volume with predicted occupancy, in which way the number of the voxels is reduced to only the occupied ones. Then, to compute the attention score of a target voxel, we define a local window centered on this voxel, within which the non-empty voxels are considered for attention computing. In this way, the computation complexity of the 3D multi-head attention can be reduced by orders of magnitude, and this module can be embedded into a network for 3D feature aggregation. Therefore, with this module, we build the first 3D transformer based top-down-bottom-up network, where a dilate-attention module and its inverse are used to downsample and upsample the 3D feature volume. In addition, to make up for the local receptive field of the sparse window attention, we add a global attention module and a global context module at the bottom of this network since the size of the volume is very small at the bottom level. With this network, the 3D shape is estimated in a coarse-to-fine manner of three levels, as is displayed in Figure 1. To the best of our knowledge, this is the first paper employing the 3D transformer for 3D scene reconstruction from a TSDF representation.

In the experiments, our method is demonstrated to outperform previous methods by a significant margin on multiple datasets. Specifically, the accuracy metric of the mesh on the ScanNet dataset is reduced by $41.8\%$, from $0.055$ to $0.032$, and the completeness metric is reduced by $25.3\%$, from $0.083$ to $0.062$. In the qualitative results, the meshes reconstructed by our method are dense, accurate, and complete. The main contributions of this work are then summarized as follows:

• We propose a sparse window multi-head attention module, with which the computation complexity of the 3D transformer is reduced significantly and becomes feasible.

• We propose a dilate-attention structure to avoid geometry degeneration in downsampling, with which we build the first top-down-bottom-up 3D transformer network for 3D feature aggregation. This network is further improved with bottom-level global attention and global context encoding.

• This 3D transformer is employed to aggregate the 3D features back-projected from the 2D features of an image sequence in a coarse-to-fine manner, and predict TSDF values for accurate and complete 3D reconstruction. This framework shows a significant improvement in multiple datasets.

## 2  RELATED WORK

**Depth-based 3D Reconstruction.** In traditional methods, reconstructing a 3D model of a scene usually involves depth estimating for a series of images, and then fusing these depths together into

a 3D data structure (Schönberger et al., 2016). After the rising of deep learning, many works have tried to estimate accurate and dense depth maps with deep neural networks (Yao et al., 2018; Wang & Shen, 2018; Chen et al., 2019; Im et al., 2019; Yuan et al., 2021; 2022; Long et al., 2021). They usually estimate the depth map of the reference image by constructing a 3D cost volume from several frames in a local window. Also, to leverage the information in the image sequence, some other methods try to propagate the message from previously predicted depths utilizing probabilistic filtering (Liu et al., 2019), Gaussian process (Hou et al., 2019a), or recurrent neural networks (Duzceker et al., 2021). Although the predicted depth maps are increasingly accurate, there is still a gap between these single-view depths and the complete 3D shape. Post mesh generation like Poisson reconstruction (Kazhdan & Hoppe, 2013), Delaunay triagulation (Labatut et al., 2009), and TSDF fusion (Newcombe et al., 2011) are proposed to solve this problem, but the inconsistency between different views is still a challenge.

**Volume-based 3D Reconstruction.** To avoid the depth estimation and fusion in 3D reconstruction, some methods try to directly regress a volumetric data structure end-to-end. SurfaceNet (Ji et al., 2017) encodes the camera parameters together with the images to predict a 3D surface occupancy volume with 3D convolutional networks. Afterward, Atlas (Murez et al., 2020) back-projects the 2D features of all images into a 3D feature volume with the estimated camera poses, and then feeds this 3D volume into a 3D U-Net to predict a TSDF volume. Then NeuralRecon (Sun et al., 2021) improves the efficiency by doing this within a local window and then fusing the prediction together using a GRU module. Recently, to improve the accuracy of the reconstruction, some methods also introduce transformers to do the fusion of 2D features from different views (Bozic et al., 2021; Stier et al., 2021). However, their transformers are all limited in 2D space and used to process 2D features, which is not straightforward in the 3D reconstruction task.

There are also some methods for object 3D shape prediction, which can infer the 3D shape of objects with only a few views (Xie et al., 2020; Wang et al., 2021a). But the network of these methods can only infer the shape of one category of small objects. Lately, some works represent the 3D shape with an implicit network, and optimize the implicit representation by neural rendering (Yariv et al., 2020; Wang et al., 2021b; Yariv et al., 2021). These methods could obtain a fine surface of an object with iterative optimization, but with the cost of a long-time reconstruction.

**Transformers in 3D Vision.** The transformer structure (Vaswani et al., 2017) has attracted a lot of attention and achieved many successes in vision tasks (Dosovitskiy et al., 2020; Liu et al., 2021). Most of them, nevertheless, are used for 2D feature extraction and aggregation. Even in 2D feature processing, the computation complexity is already quite high, so many works are proposed to reduce the resource-consuming (Dosovitskiy et al., 2020; Liu et al., 2021). Directly extending the transformer from 2D to 3D would cause catastrophic computation. Thus most works are only carefully performed on resource-saving feature extraction, e.g., the one-off straightforward feature mapping without any downsampling or upsampling (Wang et al., 2021a), where the size of the feature volume remains unchanged, or the top-down tasks with only downsampling (Mao et al., 2021), where the size of the feature volume is reduced gradually. In 3D reconstruction, however, a top-down-bottom-up structure is more reasonable for feature extraction and shape generation, as in most of the 3D-CNN-based structures (Murez et al., 2020; Sun et al., 2021; Stier et al., 2021). So in this work, we design the first 3D transformer based top-down-bottom-up structure for improving the quality of 3D reconstruction. In addition, a sparse window multi-head attention mechanism is proposed to save the computation cost. Although the sparse structure can handle the highly-sparse data, like the object detection of Lidar points (Mao et al., 2021), it is not suitable for processing a relatively-dense data, like a mesh of an indoor scene. Therefore, a sparse window structure is needed in 3D scene reconstruction, where a dense surface within a window could be sufficiently aggregated.

## 3 METHOD

### 3.1 OVERVIEW

The overview framework of our method is illustrated in Figure 1. Given a sequence of images $\{\mathbf{I}_i\}_{i=1}^N$ of a scene and the corresponding camera intrinsics $\{\mathbf{K}_i\}_{i=1}^N$ and extrinsics $\{\mathbf{P}_i\}_{i=1}^N$, we first extract the image features $\{\mathbf{F}_i\}_{i=1}^N$ in 2D space in three levels, and then back project these 2D features to 3D space, which are fused to three feature volumes in the coarse, medium, and fine levels, respectively. Afterward, these three feature volumes are aggregated by our SDF 3D transformer in

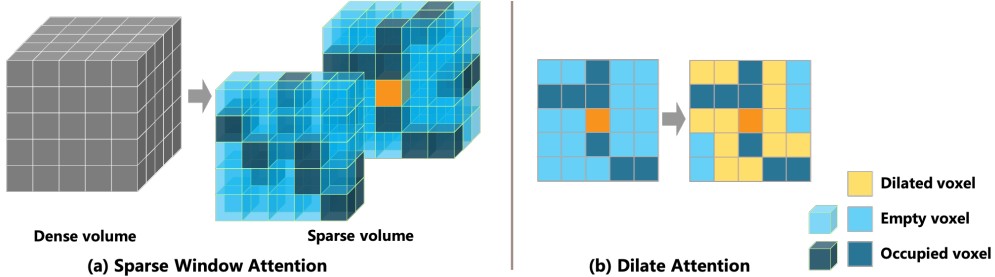

Figure 2: (a) Illustration of the sparse window attention. For calculating the attention of the current voxel (in orange), we first sparsify the volume using the occupancy prediction from the coarser level, and then search the occupied voxels (in dark blue) within a small window. The attention is hence computed based on only these neighbor occupied voxels. (b) Illustration of the dilate-attention in a 2D slice. We dilate the occupied voxels and calculate the attention of these dilated voxels (in yellow) to maintain the geometry structure.

a coarse-to-fine manner. At the coarse and medium levels, the output of the 3D transformer is two occupancy volumes $\mathbf{O^2}, \mathbf{O^1}$, while at the fine level, the output is the predicted TSDF volume $\mathbf{S^0}$. The coarse occupancy volume $\mathbf{O^2}$ and the medium occupancy volume $\mathbf{O^1}$ store the occupancy values $o \in [0, 1]$ of the voxels, which are used to sparsify the finer level. Therefore, the feature volumes could be processed sparsely to reduce the computation complexity. Finally, the predicted mesh is extracted using marching cubes (Lorensen & Cline, 1987) from the TSDF volume $\mathbf{S^0}$.

## 3.2 FEATURE VOLUME CONSTRUCTION

The 2D features $\{\mathbf{F}_i^l\}_{i=1}^N$ in three levels $l = 0, 1, 2$ are extracted by a feature pyramid network (Lin et al., 2017) with the MnasNet-B1 (Tan et al., 2019) as the backbone. The resolution of the features at these three levels are $\frac{1}{4}, \frac{1}{8}, \frac{1}{16}$, respectively. Then following Murez et al. (2020), we back project the 2D features to 3D space with the camera parameters $\{\mathbf{K}_i\}_{i=1}^N$ and $\{\mathbf{P}_i\}_{i=1}^N$, generating 3D feature volumes $\{\mathbf{V}_i^l\}_{i=1}^N$ of size $N_X \times N_Y \times N_Z$.

In previous work, usually the fusion of these feature volumes from different views is computed by taking the average (Murez et al., 2020; Sun et al., 2021). However, the back-projected features from different views contribute differently to the 3D shape, e.g., the view with a bad viewing angle and the voxels far from the surface. Therefore, a weighted average is more reasonable than taking the average. To compute these weights, for each voxel we calculate the variance of the features of different views by

$$\mathbf{Var}_i^l = (\mathbf{V}_i^l - \overline{\mathbf{V}}^l)^2, \tag{1}$$

where $\overline{\mathbf{V}}^l$ is the average of the features of all views. Then we feed the features and the variance into a small MLP to calculate the weights $\mathbf{W}_i$, which are used to compute a weighted average of the features from different views as

$$\mathbf{V}_w^l = \frac{1}{N} \sum_i \mathbf{V}_i^l \times \text{SoftMax}(\mathbf{W}_i), \tag{2}$$

where $\times$ denotes element-wise multiplication.

Inspired by Yao et al. (2018), we also calculate the total variance of all feature volumes and then concatenate it with the weighted average to the final feature volumes, as

$$\mathbf{V}^l = \{\mathbf{V}_w^l, \frac{1}{N} \sum_i \mathbf{Var}_i^l\}, \tag{3}$$

## 3.3 SPARSE WINDOW MULTI-HEAD ATTENTION

The multi-head attention structure has been shown to be effective in many vision tasks (Dosovitskiy et al., 2020; Liu et al., 2021). Most of them, however, are limited to 2D feature processing rather than 3D feature processing. This is because the computation complexity of the multi-head attention is usually higher than convolutional networks, which problem is further enlarged in 3D features. To compute this for a 3D feature volume, the attentions between a voxel and any other voxels need to be computed, i.e., $N_X \times N_Y \times N_Z$ attentions for one voxel and $N_X \times N_Y \times N_Z \times N_X \times N_Y \times N_Z$ attentions for all voxels, which is extremely large and hard to be realized in regular GPUs.

To deal with this problem and make the multi-head attention of 3D volumes feasible, we propose to use a sparse window structure to calculate the attention. As is displayed in Figure 1, in the medium and the fine level, we sparsify the volumes using the occupancy prediction $\mathbf{O^2}, \mathbf{O^1}$, and only compute the attention of the non-empty voxels. In addition, considering that the nearby voxels contribute more to the shape of the current voxel and the distant voxels contribute less, we only calculate the attention within a local window of each voxel, as is shown in Figure 2. Therefore, we are able to only calculate the multi-head attention of the occupied voxels within a small window, in which way the computation complexity is reduced significantly.

Specifically, for any non-empty voxel $v_i$ in the feature volume $V$, we first search all non-empty voxels within a $n \times n \times n$ window centered on this voxel and get the neighbor voxels $\{v_j, j \in \Omega(i)\}$. Then the query, key, and value embeddings are calculated as

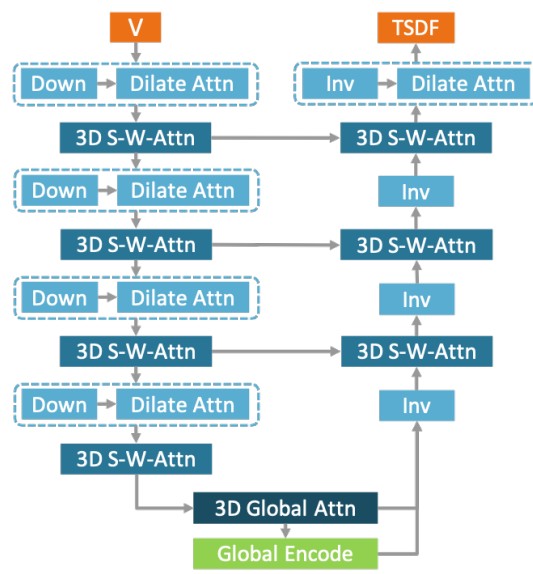

Figure 3: The structure of the SDF transformer. "S-W-Attn" denotes sparse window attention.

$$Q_i = \mathcal{L}_q(V(v_i)), K_j = \mathcal{L}_k(V(v_j)), V_j = \mathcal{L}_v(V(v_j)), \tag{4}$$

where $\mathcal{L}_q, \mathcal{L}_k, \mathcal{L}_v$ are the linear projection layers.

For the position embedding $P$, we hope to block the influence from the scale of the 3D world coordinates. Hence we compute it based on the relative voxel position in the volume rather than based on the real-world coordinates (Mao et al., 2021), as

$$P_j = \mathcal{L}_p(v_j - v_i). \tag{5}$$

Then the attention is calculated as

$$\text{Attention}(v_i) = \sum_{j \in \Omega(i)} \text{SoftMax}(Q_i(K_j + P_j)/\sqrt{d})(V_j + P_j). \tag{6}$$

In this case, the computation complexity is reduced from

$$\mathbb{O}_{\text{3D-Attn}} = N_X \times N_Y \times N_Z \times N_X \times N_Y \times N_Z \times \mathbb{O}(ij), \tag{7}$$

to

$$\mathbb{O}_{\text{SW-3D-Attn}} = N_{\text{occu}} \times n_{\text{occu}} \times \mathbb{O}(ij), \tag{8}$$

where $\mathbb{O}(ij)$ is the complexity of one attention computation between voxel $v_i$ and $v_j$, $N_{\text{occu}}$ is the number of occupied voxels in the volume, and $n_{\text{occu}}$ is the number of occupied voxels within the local window. Assuming that the occupancy rate of the volume is $10\%$ and the window size is $\frac{1}{10}$ of the volume size, the computation complexity of the sparse window attention would be only $\frac{n^3/10}{10 N_X N_Y N_Z} = \frac{1}{100000}$ of the dense 3D attention.

## 3.4 SDF 3D Transformer

Limited by the high resource-consuming of the multi-head attention, most of the previous works related to 3D transformers are only carefully performed on resource-saving feature processing, e.g., the one-off straightforward feature mapping without any downsampling or upsampling (Wang et al., 2021a), where the size of feature volumes remains unchanged, or the top-down tasks with only downsampling (Mao et al., 2021), where the size of feature volumes is reduced gradually. In 3D reconstruction, however, a top-down-bottom-up structure is more reasonable for feature extraction and prediction generation, as in most of the 3D-CNN-based structures (Murez et al., 2020; Sun et al., 2021; Stier et al., 2021). So in this work, we design the first 3D transformer based top-down-bottom-up structure, as is shown in Figure 3.

| Baseline | + SDF Transformer | + Post Dilate Attention | Ground Truth |

Figure 4: Ablation study on the ScanNet dataset.

Taking the network for the fine volume ($V^0$ in Figure 1) as an example, there are four feature levels in total, i.e. $\frac{1}{2}, \frac{1}{4}, \frac{1}{8}, \frac{1}{16}$, as shown in Figure 3. In the encoder part, at each level, a combination of downsampling and dilate-attention is proposed to downsample the feature volume. Then two blocks of the sparse window multi-head attention are used to aggregate the feature volumes. At the bottom level, a global attention block is employed to make up the small receptive field of the window attention, and a global context encoding block is utilized to extract the global information. In the decoder part, we use the inverse sparse 3D CNN to upsample the feature volume, i.e., we store the mapping of the down flow and now restore the spatial structure by inversing the sparse 3D CNN in the dilate-attention. Therefore, the final shape after the up flow should be the same as the input. Similar to FPN (Lin et al., 2017), the features in the down flow are also added to the upsampled features in the corresponding level. To enable the deformation ability, a post-dilate-attention block is equipped after the down-up flow. Finally, a submanifold 3D CNN head with Tanh activation is appended to output the TSDF prediction. For the coarse volume $V^2$ and medium volume $V^1$, two and three-level of similar structures with Sigmoid activation are adopted.

**Dilate-attention.** The direct downsampling of a sparse structure is prone to losing geometry structure. To deal with this, between each level we first downsample the feature volume, and then dilate the volume with a sparse 3D CNN with the kernel size of 3, which calculates the output if any voxel within its kernel is non-empty. The dilation operation alone may also harm the geometry, since it may add some wrong voxels into the sparse structure. Thus we calculate the sparse window attention of the dilated voxels, such that the voxels far from the surface would get low scores and do not contribute to the final shape. The dilated voxels are then joined to the downsampled volume by concatenating the voxels together. With this dilate-attention module, the 3D shape is prevented from collapsing. Without this module, the network performs badly and only generates a degraded shape.

**Global attention and global context encoding.** Since the attention blocks in the top-down flow are all local-window based, there could be a lack of the global receptive field. Considering the resolution of the bottom level is not high, we equip with a global attention block at the bottom level, i.e., we calculate the attention between each non-empty voxel and any other non-empty voxel in the volume. This could build the long-range dependency missing in the sparse window attention blocks. In addition, we use the multi-scale global averaging pooling (Zhao et al., 2017) of scales $1, 2, 3$ to extract the global context code of the scene. This encoding module could aggregate the global information and explain the illumination, global texture, and global geometry style.

## 3.5 Loss Function

The final TSDF prediction $\mathbf{S}^0$ is supervised by the log L1 distance between the prediction and the ground truth as $L^0 = |\log \mathbf{S}^0 - \log \widehat{\mathbf{S}}|$.

To supervise the occupancy predictions $\mathbf{O}^2, \mathbf{O}^1$ in the coarse and medium levels, we generate the occupancy volumes based on the TSDF values. Specifically, the voxels with TSDF of $-1 \sim 1$ are regarded as occupied, and the values are set to $1$, otherwise set to $0$. Then a binary cross-entropy loss is calculated between the prediction and the ground truth as: $L^l = -\widehat{\mathbf{O}^l} \log \mathbf{O}^l, \; l = 1, 2$.

To supervise the averaging weights $\mathbf{W}_i^l$, we use the occupancy in the back-projection following Stier et al. (2021). Intuitively, when the feature is back-projected from a 2D image to the 3D space along the camera ray using multiple depth values, we hope the voxels close to the mesh surface have bigger weights in the fusion. Therefore, the 3D position is regarded as occupied if the difference between the project depth and the true depth from the depth map is smaller than the TSDF truncation distance. Then the cross entropy loss is applied to the weights and the occupancy:

$$L_w^l = -\widehat{\mathbf{O}_i^l} \log \sigma(\mathbf{W}_i^l), \; l = 1, 2, 3, \tag{9}$$

where $\sigma$ denotes Sigmoid, and $\widehat{\mathbf{O}_i^l}$ is the ground truth occupancy in the back-projection of image $I_i$.

| Method | Acc ↓ | Comp ↓ | Chamfer ↓ | Prec ↑ | Recall ↑ | F-score ↑ |
|---|---|---|---|---|---|---|
| DeepVideoMVS (Duzceker et al., 2021) | 0.079 | 0.133 | 0.106 | 0.521 | 0.454 | 0.474 |
| Atlas (Murez et al., 2020) | 0.068 | 0.098 | 0.083 | 0.640 | 0.539 | 0.583 |
| NeuralRecon (Sun et al., 2021) | 0.054 | 0.128 | 0.091 | 0.684 | 0.479 | 0.562 |
| VoRTX (Stier et al., 2021) | 0.054 | 0.090 | 0.072 | 0.708 | 0.588 | 0.641 |
| Ours | **0.049** | **0.068** | **0.058** | **0.754** | **0.664** | **0.705** |
| Colmap (Schönberger et al., 2016) | 0.102 | 0.119 | 0.111 | 0.509 | 0.474 | 0.489 |
| MVDepthNet (Wang & Shen, 2018) | 0.129 | 0.083 | 0.106 | 0.443 | 0.487 | 0.460 |
| GP-MVS (Hou et al., 2019a) | 0.129 | 0.080 | 0.105 | 0.453 | 0.510 | 0.477 |
| DPSNet (Im et al., 2019) | 0.119 | 0.076 | 0.098 | 0.474 | 0.519 | 0.492 |
| ESTDepth (Long et al., 2021) | 0.127 | 0.075 | 0.101 | 0.456 | 0.542 | 0.491 |
| DeepVideoMVS (Duzceker et al., 2021) | 0.107 | 0.069 | 0.088 | 0.541 | 0.592 | 0.563 |
| Atlas (Murez et al., 2020) | 0.072 | 0.076 | 0.074 | 0.675 | 0.605 | 0.636 |
| NeuralRecon (Sun et al., 2021) | 0.051 | 0.091 | 0.071 | 0.630 | 0.612 | 0.619 |
| TransformerFusion (Bozic et al., 2021) | 0.055 | 0.083 | 0.069 | 0.728 | 0.600 | 0.655 |
| Ours | **0.032** | **0.062** | **0.047** | **0.829** | **0.694** | **0.754** |

Table 1: Evaluation of the 3D meshes on ScanNet. The upper part follows the evaluation in Sun et al. (2021) while the lower part follows Bozic et al. (2021). The metric definitions are explained in the appendix.

| Method | Abs Rel ↓ | Abs Diff ↓ | Sq Rel ↓ | RMSE ↓ | $\delta\text{-}1.25$ ↑ | $\delta\text{-}1.25^2$ ↑ | $\delta\text{-}1.25^3$ ↑ |
|---|---|---|---|---|---|---|---|
| Colmap (Schönberger et al., 2016) | 0.137 | 0.264 | 0.138 | 0.502 | 0.834 | − | − |
| MVDepthNet (Wang & Shen, 2018) | 0.098 | 0.191 | 0.061 | 0.293 | 0.896 | 0.977 | 0.994 |
| GP-MVS (Hou et al., 2019a) | 0.130 | 0.239 | 0.339 | 0.472 | 0.906 | 0.967 | 0.980 |
| DPSNet (Im et al., 2019) | 0.087 | 0.158 | 0.035 | 0.232 | 0.925 | 0.984 | 0.995 |
| Atlas (Murez et al., 2020) | 0.065 | 0.124 | 0.043 | 0.251 | 0.936 | 0.971 | 0.986 |
| NeuralRecon (Sun et al., 2021) | 0.065 | 0.106 | **0.031** | **0.195** | 0.948 | 0.961 | 0.975 |
| Vortx (Stier et al., 2021) | 0.061 | 0.096 | 0.038 | 0.205 | 0.943 | 0.973 | 0.987 |
| Ours | **0.051** | **0.086** | 0.033 | 0.199 | **0.958** | **0.980** | **0.990** |

Table 2: Evaluation of the 2D depth maps on the ScanNet dataset. The upper part shows the results of depth-based methods, while the lower part shows volumetric methods, whose depths are rendered from the meshes.

# 4 EXPERIMENTS

## 4.1 EXPERIMENTS SETUP

Our work is implemented in Pytorch and trained on Nvidia V100 GPUs. The network is optimized with the Adam optimizer ($\beta_1 = 0.9, \beta_2 = 0.999$) with learning rate of $1 \times 10^{-4}$. For a fair comparison with previous methods, the voxel size of the fine level is set to 4cm, and the TSDF truncation distance is set to triple the voxel size. Thus the voxel size of the medium and the coarse levels are 8 cm and 16 cm, respectively. For the balance of efficiency and receptive field, the window size of the sparse window attention is set to 10. For the view selection, we first follow Hou et al. (2019b) to remove the redundant views, i.e., a new incoming frame is added to the system only if its relative translation is greater than 0.1 m and the relative rotation angle is greater than 15 degree. Then if the number of the remaining views exceeds the upper limit, a random selection is adopted for memory efficiency. The view limit is set to 20 in the training, which means twenty images are input to the network for one iteration, while the limit for testing is set to 150. Our framework runs at an online speed of 75 FPS for the keyframes. Detailed efficiency experiments are reported in the supplemental materials.

ScanNet (Dai et al., 2017) is a large-scale indoor dataset composed of 1613 RGB-D videos of 806 indoor scenes. We follow the official train/test split, where there are 1513 scans used for training and 100 scans used for testing. TUM-RGBD (Sturm et al., 2012) and ICL-NUIM (Handa et al., 2014) are also two datasets composed of RGB-D videos but with small-number scenes. Therefore, following previous methods (Stier et al., 2021), we only perform the generalization evaluation of the model trained on ScanNet on these two datasets, where 13 scenes of TUM-RGBD and 8 scenes of ICL-NUIM are used.

## 4.2 EVALUATION

To compare with previous methods, we evaluate the proposed method on the ScanNet test set. The quantitative results are presented in Table 1 and the qualitative comparison are displayed in Figure 5.

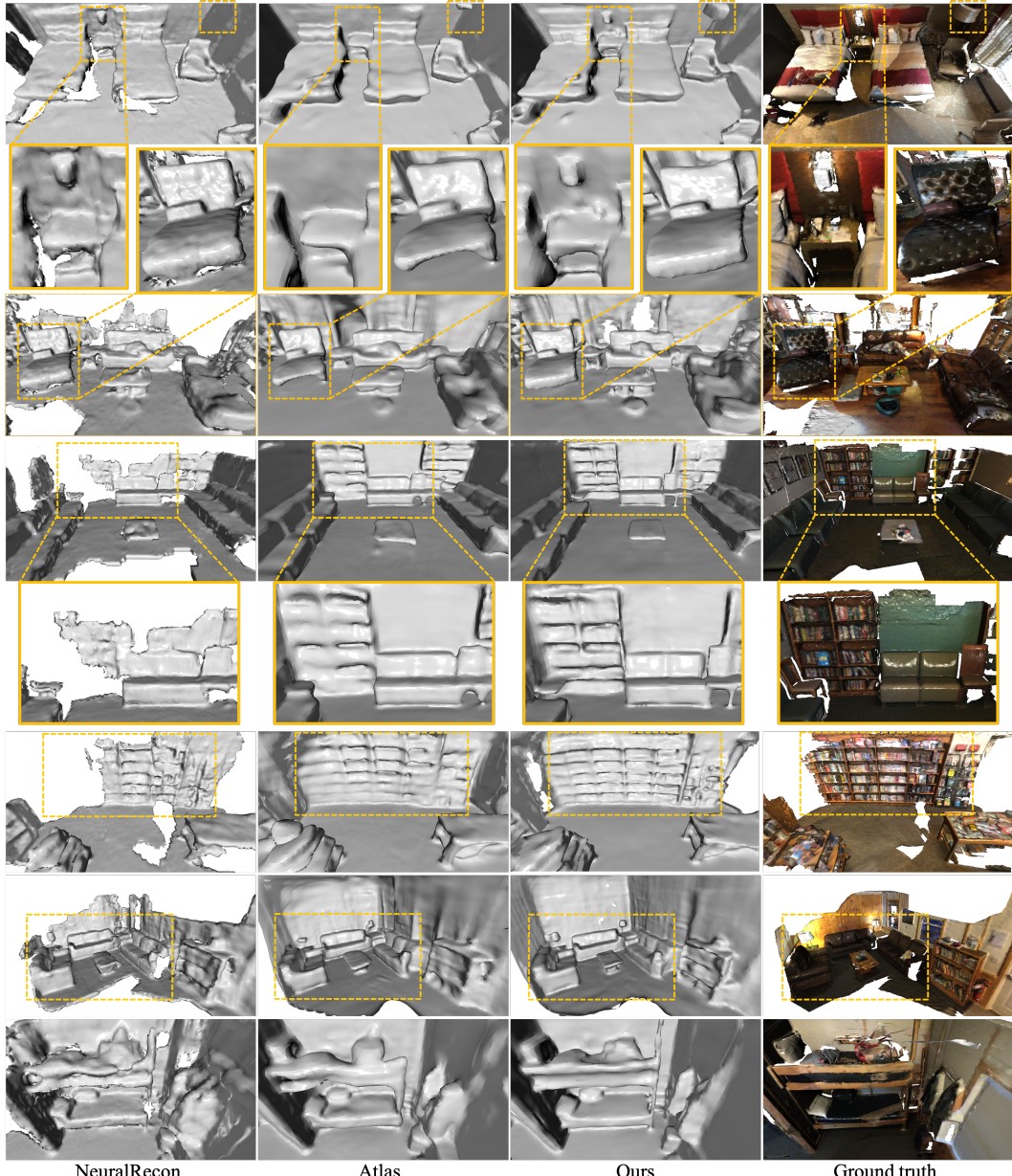

| NeuralRecon | Atlas | Ours | Ground truth |

Figure 5: The qualitative results on the ScanNet dataset. Texture-less rendering is displayed in the appendix.

We first directly evaluate the reconstructed meshes with the ground-truth meshes, and obtain a significant improvement from previous methods, improving from F-score = 0.641 to F-score = 0.705, as shown in Table 1. Then following Bozic et al. (2021), we add the same occlusion mask at evaluation to avoid penalizing a more complete reconstruction, which is because the ground-truth meshes are incomplete due to unobserved and occluded regions, while our method could reconstruct a more complete 3D shape, as shown in Figure 5. This results in a more reasonable evaluation, as in the second part of Table 1. The improvement is further enlarged, from F-score = 0.655 to F-score = 0.754 compared to previous best method. The accuracy error is decreased from 0.055 m to 0.032 m, which is almost half (41.8%) of the previous best method, while the completeness error is decreased by 25.3%, from 0.083 m to 0.062 m. This owes to the feature aggregating ability of the proposed 3D SDF transformer, which can predict a more accurate 3D shape. This is also demonstrated in the generalization experiments on ICL-NUIM and TUM-RGBD datasets, as shown in Figure 3.

After evaluating the reconstructed meshes, we also evaluate the depth accuracy of our method. Since our method does not predict the depth maps explicitly, we render the predicted 3D shape to the

image planes and get the depth maps, following previous methods (Murez et al., 2020). The results are shown in Table 2, from which we can see our method decreases the error a lot from previous methods. The relative error is reduced by $16.4\%$, from $0.061$ to $0.051$. The accuracy of the depth maps also demonstrates the accurate feature analysis ability of the proposed 3D SDF transformer.

From the qualitative visualization in Figure 5, we can see our method can predict a complete and accurate 3D shape. Previous methods which can recover a complete mesh usually reconstruct a smooth 3D shape with losing some details (Murez et al., 2020). However, our method could predict a more complete mesh than the ground truth, while the details of the 3D shapes are better recovered. Please note that for a fair comparison, the voxel size is set to $4$ cm, such that it is hard to reconstruct the geometry details less than $4$ cm.

| | Method | Acc ↓ | Comp ↓ | Prec ↑ | Recall ↑ | F-score ↑ |
|---|---|---|---|---|---|---|
| ICL | Atlas | 0.175 | 0.314 | 0.280 | 0.194 | 0.229 |
| | NeuralRecon | 0.215 | 1.031 | 0.214 | 0.036 | 0.058 |
| | VoRTX | 0.102 | 0.146 | 0.449 | 0.375 | 0.408 |
| | Ours | **0.083** | **0.142** | **0.522** | **0.390** | **0.447** |
| TUM | Atlas | 0.208 | 2.344 | 0.360 | 0.089 | 0.132 |
| | NeuralRecon | 0.130 | 2.528 | 0.382 | 0.075 | 0.115 |
| | Vortx | 0.175 | **0.314** | 0.280 | **0.194** | 0.229 |
| | Ours | **0.129** | 0.455 | **0.406** | 0.173 | **0.254** |

Table 3: Generalization experiments on the ICL-NUIM and TUM-RGBD datasets.

### 4.3 ABLATION STUDY

**SDF transformer.** To verify the effectiveness of the proposed SDF transformer, we first build a baseline model with the same structure as Figure 1, but the 3D SDF transformer is replaced by a UNet structure of 3D CNN. Adding the variance fusion would improve the mesh in some clutter areas and slightly increase the performance. Then we add a base version of the SDF transformer, which does not include the global module and the post-

| Method | Acc ↓ | Comp ↓ | Prec ↑ | Recall ↑ | F-score ↑ |
|---|---|---|---|---|---|
| Baseline | 0.056 | 0.089 | 0.698 | 0.587 | 0.636 |
| + Var Fusion | 0.054 | 0.090 | 0.713 | 0.594 | 0.647 |
| + SDF Former | 0.036 | 0.065 | 0.807 | 0.671 | 0.732 |
| + Global | 0.033 | 0.064 | 0.823 | 0.676 | 0.741 |
| + Post-Dila-Attn | 0.032 | 0.062 | 0.829 | 0.694 | 0.754 |
| Window Size 1 | 0.052 | 0.086 | 0.721 | 0.604 | 0.656 |
| 3 | 0.044 | 0.078 | 0.768 | 0.636 | 0.695 |
| 5 | 0.037 | 0.069 | 0.799 | 0.660 | 0.730 |
| 8 | 0.033 | 0.065 | 0.822 | 0.682 | 0.746 |
| 10 | 0.032 | 0.062 | 0.829 | 0.694 | 0.754 |

Table 4: Ablation study on the ScanNet dataset. Components are added one by one in the upper part.

dilate-attention module. The performance is significantly improved with this module, as is shown in Table 4 and Figure 4. The reconstructed meshes possess much more geometry details compared to the baseline.

**Global module.** We next add the global module, including the bottom-level global attention and the global context code. The sparse window attention block can only obtain the long-range dependency within a local window. Thus it may have problems when it can not get enough information within this local window, e.g., the texture-free regions. Also, the global module could reason the global information like the illumination and the texture style.

**Dilate attention.** The dilate attention module is crucial in the SDF transformer, so we can not remove all the dilate attention blocks. That will destroy the whole framework and generate a degraded 3D shape. Therefore, we only ablate the post dilate attention block after the down-up flow. This block could deform the shape and make it more complete, e.g., making up the crack as shown in Figure 4. From the quantitative results in Table 4, we can also see the improvement of completeness.

**Window size.** As shown in Table 4, we study the impact of the window size of the attention. It is expected that a larger window size would generate a better result, since the range of the dependency is longer, but with the cost of more resource consumption. We choose 10 as the default size, considering that the performance improvement is minor after that.

## 5 CONCLUSION

We propose the first top-down-bottom-up 3D transformer for 3D scene reconstruction. A sparse window attention module is proposed to reduce the computation, a dilate attention module is proposed to avoid geometry degeneration, and a global module at the bottom level is employed to extract the global information. This structure could be used to aggregate any 3D feature volume, thus it could be applied to more 3D tasks in the future, such as 3D segmentation.

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

# A APPENDIX

## A.1 MORE DETAILS

**Volume sparsify.** The ground-truth occupancy volumes are generated based on the ground-truth TSDF volumes. The voxels with TSDF value of $[-1, 1]$ are regarded as occupied and set to $1$, otherwise set to $0$. In the training or the inference, after the occupancy volume is predicted in the coarser level, the voxels of occupancy value less than $0.5$ are regarded as empty and discarded, while the remaining voxels are regarded as occupied and transmitted to the next level. The sparse volume is stored in a hash table, where the key of the table is the hash value of the voxel, and the value of the table stores the corresponding feature. In the coarsest level, all voxels are regarded as non-empty and stored in the hash table, which does not consume much memory because the size of the volume is small.

**Training and inference.** The training and inference are performed in a similar way. For a given sequence of images, first a view selection is performed to select images with translation greater than 0.1m and rotation greater than 15 degrees. Then a random selection is adopted from the remaining images if the number exceeds the upper limit. These images are then fed to the 2D backbone for feature extraction, after which the features are fused to a 3D volume and fed to the 3D part to produce the TSDF volume prediction. The final mesh is extracted by marching cubes from the TSDF volume. This process is the same as previous methods like Atlas, TransformerFusion or VORTX.

For the number of the upper limit of the images, actually any number for the sequence length is okay for our framework, although more images lead to a better reconstruction of a scene. In our experiments, the number for training is set to 20 and the number for inference is set to 150.

| Method | Per Frame Time | Per Scene Time | FPS |
|---|---|---|---|
| Atlas (Murez et al., 2020) | 71 ms | 840 ms | 14 |
| NeuralRecon (Sun et al., 2021) | 30 ms | 0 ms | 33 |
| TransformerFusion (Bozic et al., 2021) | 130.5 ms | 243.3 ms | 7 |
| VoRTX (Stier et al., 2021) | 71.4 ms | 231.7 ms | 14 |
| Ours | 13.3 ms | 286.4 ms | 75 |

Table 5: Efficiency experiments.

## A.2 EFFICIENCY

The runtime analysis is presented in Table 5. For a fair comparison to previous methods, the time is tested on a chunk of size $1.5 \times 1.5 \times 1.5$ m$^3$ with an Nvidia RTX 3090 GPU. Our framework consists of two parts: one is the per-frame part, including the feature extraction of the 2D images; the other one is the per-scene part, including the feature fusion, 3D feature processing, and mesh generation. The per-frame model runs for every keyframe, i.e., it keeps running whenever a new keyframe comes. Differently, the per-scene model runs only once for generating a mesh reconstruction of a scene, i.e., it only works after all frames are fed, or when we need to output a mesh. Therefore, the online speed of a normal running is 75 FPS, which only performs the mesh generation once at the end.

## A.3 METRICS

The definitions of the 2D metrics and 3D metrics used for evaluation are explained in Table 6.

## A.4 LIMITATIONS

Due to the volume representation, our framework is limited by the trade-off between the resolution of the volume and the memory consumption. A smaller voxel size would cost much more memory. The voxel size is set to $4$ cm, such that the geometry details less than $4$ cm are hard to be recovered.

| Metrics | Definition |
|---|---|
| Abs Rel | $\frac{1}{n}\sum |d - d^*|/d^*$ |
| Abs Diff | $\frac{1}{n}\sum |d - d^*|$ |
| Sq Rel | $\frac{1}{n}\sum |d - d^*|^2/d^*$ |
| RMSE | $\sqrt{\frac{1}{n}\sum |d - d^*|^2}$ |
| $\delta - 1.25^i$ | $\frac{1}{n}\sum (\max(\frac{d}{d^*}, \frac{d^*}{d}) < 1.25^i)$ |
| Acc | $\mathrm{mean}_{p \in P}(\min_{p^* \in P^*} ||p - p^*||)$ |
| Comp | $\mathrm{mean}_{p^* \in P^*}(\min_{p \in} ||p - p^*||)$ |
| Chamfer distance | $\frac{\mathrm{Acc} + \mathrm{Comp}}{2}$ |
| Prec | $\mathrm{mean}_{p \in P}(\min_{p^* \in P^*} ||p - p^*|| < 0.05)$ |
| Recall | $\mathrm{mean}_{p^* \in P^*}(\min_{p \in} ||p - p^*|| < 0.05)$ |
| F-score | $\frac{2 \times \mathrm{Prec} \times \mathrm{Recall}}{\mathrm{Prec} + \mathrm{Recall}}$ |

Table 6: Metric definitions. $n$ denotes the number of pixels with both valid ground truth and prediction, $d$ and $d^*$ denote the predicted and the ground-truth depths, $p$ and $p^*$ denote the predicted and the ground-truth point clouds.

## A.5 ROBUSTNESS TO THE POSE NOISE

Our method is based on the given accurate camera poses, which is the same as previous state-of-the-art methods like Atlas (Murez et al., 2020), NeuralRecon (Sun et al., 2021), and Transformer-Fusion (Bozic et al., 2021), where the camera poses are obtained by the standard SfM or SLAM systems. To inspect the robustness of our method to the pose errors, we add the Gaussian noise to the camera poses. A translation noise $[N_x, N_y, N_z]$ of $N = Gauss\{0, \sigma_T\}$ is added to the translation of the pose, while a rotation noise $[N_{roll}, N_{pitch}, N_{yaw}]$ of $N = Gauss\{0, \sigma_R\}$ is added to the three angles of the pose. The metrics following NeuralRecon (Sun et al., 2021) are reported in Table 7. From the results, we can see our system can handle some translation errors but cannot handle the rotation errors well. But if the poses of only some frames are miscalculated, e.g., 10% of all frames, the performance decrease would be under control.

| Ratio | $\sigma_T$ (cm) | $\sigma_R$ (deg) | Acc ↓ | Comp ↓ | Prec ↑ | Recall ↑ | F-score ↑ |
|---|---|---|---|---|---|---|---|
| 0 | 0 | 0 | 0.049 | 0.068 | 0.754 | 0.664 | 0.705 |
| 100% | 0.5 | 0 | 0.050 | 0.068 | 0.745 | 0.658 | 0.698 |
| 100% | 1 | 0 | 0.055 | 0.073 | 0.708 | 0.626 | 0.663 |
| 100% | 0 | 0.5 | 0.084 | 0.117 | 0.525 | 0.446 | 0.480 |
| 100% | 0 | 1 | 0.109 | 0.185 | 0.406 | 0.314 | 0.351 |
| 100% | 0.5 | 0.5 | 0.084 | 0.117 | 0.516 | 0.435 | 0.471 |
| 100% | 1 | 1 | 0.114 | 0.187 | 0.380 | 0.296 | 0.330 |
| 10% | 0.5 | 0.5 | 0.055 | 0.074 | 0.715 | 0.629 | 0.668 |
| 10% | 1 | 1 | 0.064 | 0.087 | 0.662 | 0.577 | 0.616 |

Table 7: Experiments with pose noise following NeuralRecon (Sun et al., 2021) metrics.

## A.6 RESULTS WITH SMALLER VOXEL SIZE

As expected, a smaller voxel size leads to a more accurate reconstruction but consumes much more GPU memory. We have trained the models with voxel sizes of 2cm and 3cm, but it is hard to evaluate the models in the large scene of ScanNet test set, because the model of 2cm requires too much memory of the GPU. Thus we only compare them on a medium scene, i.e., Scene-709, as reported in Table 8. The per-frame time remains unchanged while the per-scene time increases.

| Voxel Size | Acc ↓ | Comp ↓ | Prec ↑ | Recall ↑ | F-score ↑ | Per Scene Time |
|------------|-------|--------|--------|----------|-----------|----------------|
| 4cm | 0.033 | 0.053 | 0.837 | 0.730 | 0.780 | 286.4 ms |
| 3cm | 0.025 | 0.061 | 0.882 | 0.756 | 0.814 | 435.6 ms |
| 2cm | 0.019 | 0.062 | 0.913 | 0.764 | 0.832 | 891.2 ms |

Table 8: Evaluation of different voxel sizes on Scene-709.

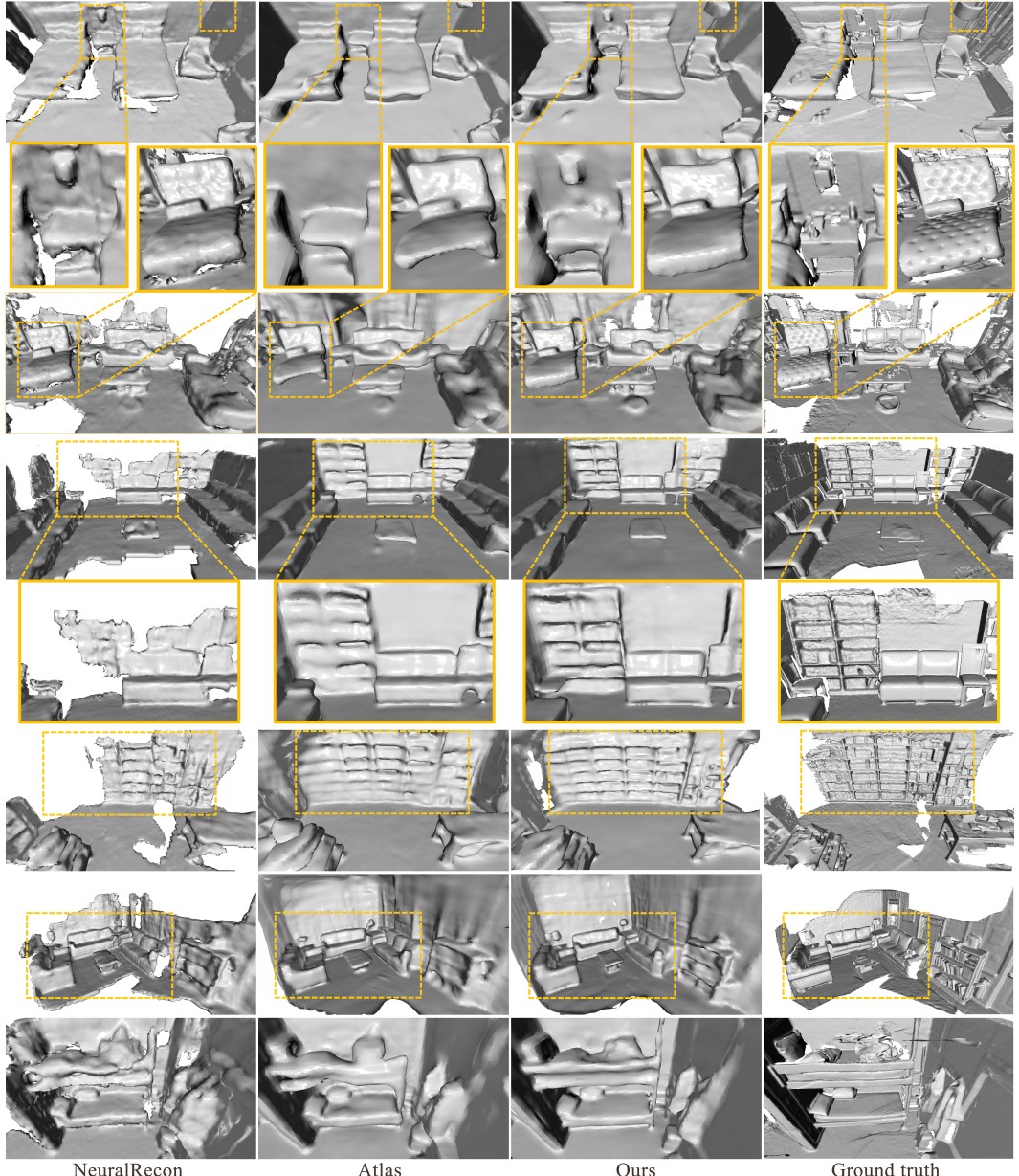

NeuralRecon    Atlas    Ours    Ground truth

Figure 6: Texture-less rendering of the ground-truth meshes for the qualitative comparison on the ScanNet dataset.

## A.7 MORE RESULTS

The texture-less rendering of the ground-truth meshes is shown in Figure 6. More results are presented in Figure 7 and Figure 8.

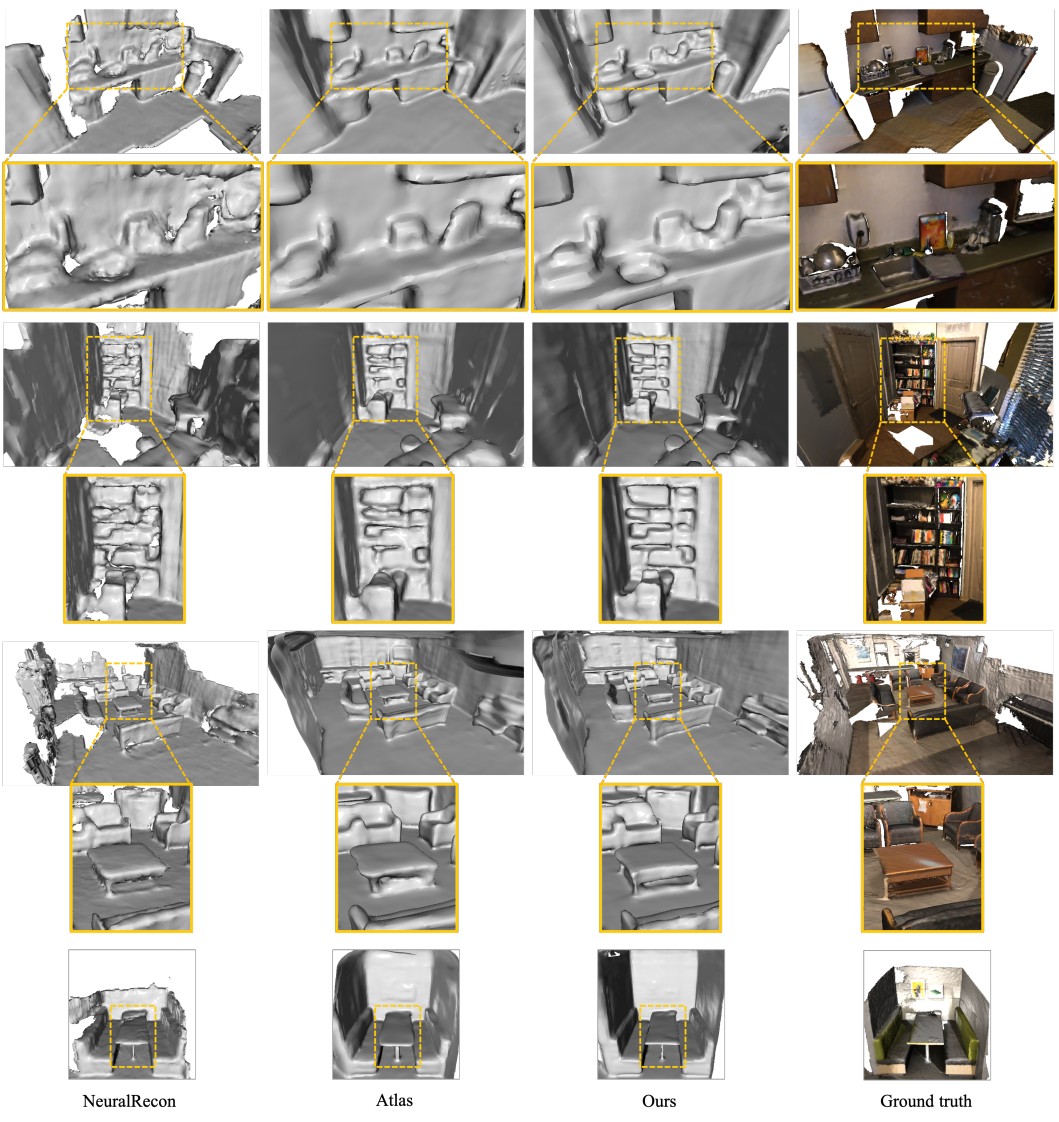

Figure 7: More qualitative results.

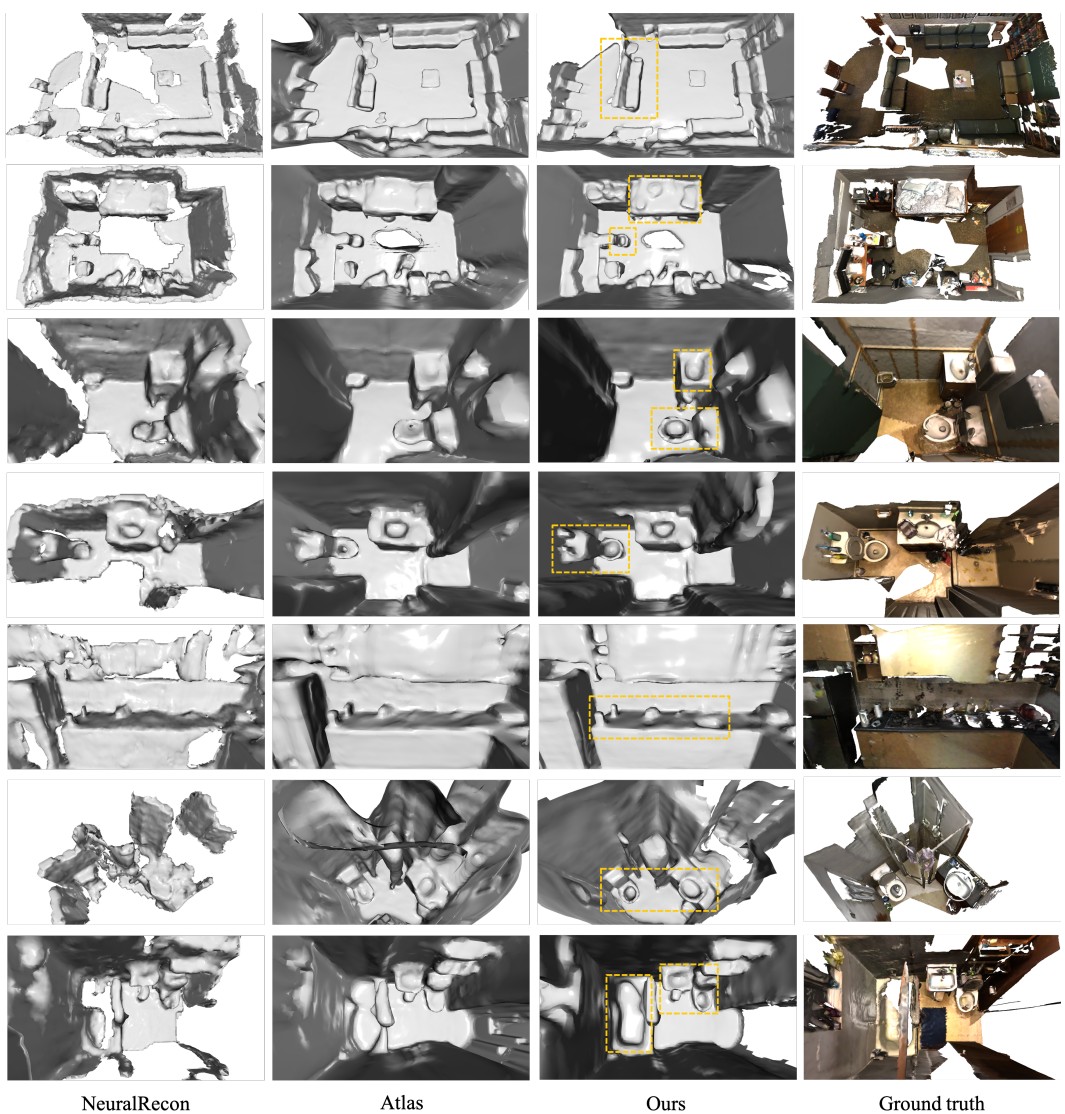

NeuralRecon        Atlas        Ours        Ground truth

Figure 8: More qualitative results.

