# OpenReview forum: "Monocular Scene Reconstruction with 3D SDF Transformers"
_ICLR.cc/2023/Conference — ICLR 2023 poster_

### Official Review · Reviewer_mQ2G · 2022-10-22

**Confidence:** 4
**Correctness:** 4
**Technical Novelty And Significance:** 3
**Empirical Novelty And Significance:** Not applicable
**Recommendation:** 5

**Clarity, Quality, Novelty And Reproducibility:**


I think the paper is clear except for the model description (unfortunately) as noted in the weaknesses.  This is unfortunate since the paper otherwise is of high quality (related work, intro, visuals, evaluation).

Because of the lack of clarity in the model description, it will be hard to reproduce the results based on the paper alone. However, the authors promise the release of the code which should make it fairly straight forward to reproduce.

**Strength And Weaknesses:**

The paper is overall well written with some problems in the model section. The illustrations are well done and help communicate the system and properties of the components. Figure 2 was important for me to understand the dilated attention (dilated can have different meanings as well!) On that note I would consider changing the name to something else. Dilation in CNNs is usually a step size between pixels/features used at the input into the kernel. But here we are growing the set of 3d occupied 3d voxels. Its a bit different.

One of the main weak points to me is the explanation of the SDF-Transformer model.
- In Fig3 there are some arrows from the left to the right side of the multi-scale transformer system at the level of the 3D S-W-Attn blocks. As far as I can tell there is no explanation of what that connection does? Does it mean we are using the feature volume on the down-sample path as the queries in the upsample path?
- Also the connections from the feature volumes after Fusion (V^i in Fig 1) are not shown in Fig3. I assume they are also arrows into the 3D S-W-Attn module on the downsample-side (left) of the diagram?
- Also I am assuming 3D S-W-Attn stands for Sparse Window attention module? Please clarify in the text.

I am also not 100% sure about the dilated attention. Specifically I am confused about this sentence:
> Then we calculate the sparse window attention of the dilated voxels and join them into the downsampled volume

How is the volume downsampled? Local averaging?  What does it mean to join them into the downsampled volume? Does it mean we are running some kind of attention mechanism? If so what are keys and values for this one? Or is the attention map inside the self attention module modified?  Or does this just mean that for the purposes of the S-W-Attn module we just append the list of dilated voxels to the list of occupied voxels?  If we dilate the attention map to extend over non-occupied voxels how do we have features in those voxels? Or is this not using a sparse feature volume?

In the feature volume construction Eq 1 is called variance but it looks more like a squared deviation from the mean? Usually the sample variance would be computed as 1/(N-1) \sum_i (V_i - V_mean)^2 ? So more like the "total variance" in Eq 3. Just wanting to double check on the notations here.


 The quantitative evaluation is well done. Comparison with major related work on the key datasets.  The ablation studies are important and clearly support the use of the 3D Transformers over 3D CNNs. Since this is the core contribution these ablations are key.

Lots of qualitative visualizations illustrate what to expect from the approach.  However, in the qualitative comparisons the GT mesh should be shown without color so we can better compare the geometric accuracy.  While I do see that the gains from larger window size seem to saturate from 8->10 I would still have liked to see some higher window sizes as well.  There is an opportunity to show the run-time tradeoff in this ablation. I.e. how much slower does the model get when we go from 8->10 or 10->12 window size? I think that would make it pretty clear why we want to stick with 10?

**Summary Of The Paper:**

The manuscript proposes to to use transformer layers applied to 3D voxel tokens at different levels of detail in order to improve the reconstruction quality of monocular 3D reconstruction (given frame poses) over related work that typically relies on 3D CNNs for multi-view aggregation and fusion.

The core contribution of the work is adapting the transformer paradigm to 3D reconstruction. This involves overcoming the cubic scaling of 3D space via sparse and windowed attention modules, and making sure the geometry does not degrade while downsampling via dilated attention.

**Summary Of The Review:**


My main concern with the paper is the clarity of the description of the 3D Transformer network that I could not completely follow in full detail.  Otherwise the paper is is solid and makes a worthwhile contribution to the community.  If the clarity problems can be resolved, I think this is a good paper that should be accepted. Until then I am leaning marginally towards reject.

---

> ### Author Response · Authors · 2022-11-18
> **Author Responses to Reviewer mQ2G**
>
> Thanks for the valuable feedback and constructive comments. In the following, we address the reviewer's comments point by point.
>
> 1. #### Different meanings of dilation.
>
>     We agree that there are multiple meanings of "dilation" as mentioned by our reviewer.
>     In this paper, "dilation" refers to the dilation operation in the morphological transformation, which means joining the neighbor pixels in.
>     We are considering changing the name but have not found a better name. Maybe "expansion"?
>     We would like to modify the confused "dilation" if our reviewer can provide some proposals.
>
> 2. #### Explanation of the SDF-Transformer model.
>
>     We are sorry for the unclear statement.
>     We have added the following detailed explanations to section 3.4 in the modified manuscript:
>     (1) The arrows mean adding the features from the down-sample path to the up-sample path, just as in the standard feature pyramid network (FPN) structure.
>     (2) As stated in the manuscript, the network in Fig 3 shows the structure at the fine level, and the structures of the coarse and medium levels are similar.
>     The feature volume after fusion ($V^i$ in Fig 1) is just the input of the network in Fig 3. The left-top block in Fig 3 denotes the connections from the feature volume $V^i$.
>     (3) Yes, S-W-Attn stands for sparse window attention module.
>
>
> 3. #### Explanation of the dilate attention.
>
>     We are sorry for the unclear statement. Yes, the down-sampling is performed by local averaging, and "joining" means appending the dilated voxels to the occupied voxels.
>     As stated in the manuscript, the dilation is realized by the sparse 3D CNN, which calculates the features of the dilated voxels based on the features of the occupied voxels within its kernel. In other words, the sparse 3D CNN will calculate the feature of an empty voxel if the kernel centered on this empty voxel contains any occupied voxel.
>     We have added the explanation above to section 3.4 in the modified manuscript.
>
> 4. #### Rename of Variance.
>
>     Thanks for the advice. The name of variance follows MVSNet [Yao et al. (2018)].
>     We would like to change the name if the reviewer have some more accurate name proposals.
>
> 5. #### Display of GT meshes without color.
>
>     Thanks for the suggestion. Following this suggestion, we have added the color-less GT meshes to Figure~6 in the appendix in the modified manuscript.
>
> 6. #### Experiments of window size.
>
>     Thanks for the advice. More results of a larger window size are reported in the Table below, where we can see that the performance increases with the rising of the window size, but saturates after the window size is larger than 10.
>     The per-frame time remains unchanged while the per-scene time increases with the rising of the window size.
>
>
>
>     | Window Size  |  Acc $\downarrow$   | Comp $\downarrow$   | Prec $\uparrow$   | Recall $\uparrow$   | F-score $\uparrow$   | Per Frame Time  | Per Scene Time  |
>     | ----------- | ----------- | ----------- | ----------- | ----------- | ----------- | ----------- |----------- |
>     | 1 | 0.052 | 0.086 | 0.721 | 0.604 | 0.656 | 13.3 ms | 132.7 ms |
>     | 3 | 0.044 | 0.078 | 0.768 | 0.636 | 0.695 | 13.3 ms | 182.1 ms |
>     | 5 | 0.037 | 0.069 | 0.799 | 0.660 | 0.730 | 13.3 ms | 238.4 ms |
>     | 8 | 0.033 | 0.065 | 0.822 | 0.682 | 0.746 | 13.3 ms | 266.9 ms |
>     |10 | 0.032 | 0.062 | 0.829 | 0.694 | 0.754 | 13.3 ms | 286.4 ms |
>     |12 | 0.032 | 0.061 | 0.827 | 0.696 | 0.755 | 13.3 ms | 305.1 ms |
>     |15 | 0.032 | 0.061 | 0.828 | 0.696 | 0.755 | 13.3 ms | 337.8 ms |
>
>     Table 1. Ablation study of the window size.

---

### Official Review · Reviewer_Zo2i · 2022-10-23

**Confidence:** 4
**Correctness:** 4
**Technical Novelty And Significance:** 3
**Empirical Novelty And Significance:** 3
**Recommendation:** 8

**Clarity, Quality, Novelty And Reproducibility:**

About Novelty please see the above section.

Clarifications:
1. It is good to show some qualitative results when dilated attention is not applied.
2. How is V^2 (the sparse volume) generated in the coarsest level (see Figure 1)? What is the density of the coarsest level volume?

**Strength And Weaknesses:**

Strengths:
1. The approach is novel. Although 1), 2) and 4) are not ideas that are hard to figure out individually, it is good to put them together to build the 3D reconstruction system, and investigate their effects to the reconstruction quality. Moreover, I think idea 3) is something novel which might inspire or followed by other 3D sparse volume UNet style work if it is effective as described in the paper.
2. The experiment is thorough. The comparison with other methods has a good coverage, and it shows a fair amount of improvement comparing to the previous arts. Also the ablation study clearly shows the benefits of each proposed component.

Weaknesses:
1. This work might not work very well with very small details due to the volume size (note that this is discussed in the appendix). However, it is actually not slow for offline reconstruction, and it would be interesting to see the results with smaller volume sizes and the analysis of the running time with regard to the volume sizes. Remember sparsity should help a lot for small volume sizes.

**Summary Of The Paper:**

This paper aims for 3D reconstruction in a coarse-to-fine manner, and the key ideas include: 1) variance based 3D volume feature fusion; 2) sparse window multi-head attention; 3) dilate attention; and 4) SDF transformer backbone for the 3D volume.

**Summary Of The Review:**

This work has good technique contributions, and the experiment is well done.

---

> ### Author Response · Authors · 2022-11-18
> **Author Responses to Reviewer Zo2i**
>
> Thanks for the valuable feedback and encouraging comments. In the following, we address the reviewer's comments point by point.
>
> 1. ####  Results with smaller voxel size.
>
>     In the following, we add the statistical experiment as suggested by our reviewer.
>     As expected, a smaller voxel size leads to a more accurate reconstruction but consumes much more GPU memory.
>     We have trained the models with voxel sizes of 2cm and 3cm, but it is hard to evaluate the models in the large scene of ScanNet test set, because the model of 2cm requires more memory than our GPU.
>     Thus we only compare them on a medium scene, i.e., Scene-709, as reported in the below Table.
>     The per-frame time remains unchanged while the per-scene time increases.
>
>
>     | Voxel Size  | Acc \downarrow   | Comp \downarrow   | Prec $\uparrow$   | Recall $\uparrow$   | F-score $\uparrow$   | Per Scene Time   |
>     | ----------- | ----------- | ----------- | ----------- | ----------- | ----------- | ----------- |
>     | 4cm | 0.033 | 0.053 | 0.837 | 0.730 | 0.780 | 286.4 ms |
>     | 3cm | 0.025 | 0.061 | 0.882 | 0.756 | 0.814 | 435.6 ms |
>     | 2cm | 0.019 | 0.062 | 0.913 | 0.764 | 0.832 | 891.2 ms |
>
>     Table 1. Evaluation of different voxel sizes on Scene-709.
>
>
>
> 2. ####  Qualitative results when dilated attention is not applied.
>
>     Thanks for the advice.
>     However, we are sorry that the dilate-attention is a critical module in our framework and it is hard to remove them all, in which case no meaningful results could be obtained. The results after removing the last dilate-attention module can be seen in Figure~4, where the leg of the chair is cracked.
>
>
> 3. ####  Density of the coarsest volume.
>
>     Sorry for the unclear statement.
>     The volume $V^2$ is actually completely dense. At the coarsest level, we just convert the dense volume to the sparse data structure without removing any voxel, since we do not have any information about the sparsity of the scene.
>     In other words, all voxels are regarded as non-empty.
>     This is fine since the volume size is small at the coarsest level.
>     We have explained it in section A.1 of the appendix due to the limited space of the main body.

---

### Official Review · Reviewer_N9G6 · 2022-10-27

**Confidence:** 4
**Correctness:** 1
**Technical Novelty And Significance:** 2
**Empirical Novelty And Significance:** 3
**Recommendation:** 6

**Clarity, Quality, Novelty And Reproducibility:**

As mentioned in the weaknesses, the experiment part lacks some details and explanations. Refer to the weaknesses.

Analysis on limitations is a must as it helps readers and engineers identify possible problems more easily. A responsible reviewer will always rate the paper higher for having it and the authors should consider that.

Although the novelty maybe sufficient, particularly as engineering sparse attention transformers is non-trivial, it is advisable to pose the paper more clearly with respect to related works.

**Strength And Weaknesses:**

Strengths:

The problem statement is well explained and has a nicely written related work section.

The illustrations and explanations show efforts and consideration. The simplicity of the figures draw the reader’s attention to the key aspects of the paper.

The proposed architecture looks reasonable and ablation experiments have been provided for the empirical justification.

The qualitative results show where the proposed approach have highest gains clearly.

Weaknesses:

I have reservations on the way the paper has been posed as an independently explored approach for SDF fusion in relation to architectures such as NeuralRecon.

Discussions on what does not work when directly integrating the transformers instead of GRU would help the reader more. Some details in the experiments are missing

Some of the writing contains rather “loosely written” phrases. Experiments writing can be improved. Minor problems exist in citations. E.g., Schonberger 2016 is not SLAM. NeuralRecon citation is missing in the experiments section.

What is the sequence length used for the experiments at a time? Are the images input one by one or it requires a certain number, say 5 or 10 images for a single iteration of training. If so what is this length?

How is the inference performed? More details should be explicitly provided on inference.

Figure 5 gt has texture which makes the comparison difficult. It is better to either have both or only the texture-less rendering for visualization.

**Summary Of The Paper:**

The paper describes a method to fuse coarse-to-fine Truncated Signed Distance Fields (TSDF) predictions over a number of time frames in an image sequence captured by monocular cameras. The TSDF are directly regressed from images with known poses at different scales, similarly to previous works such as NeuralRecon. The main contribution here is to introduce a transformer attention (e.g., instead of a GRU) module to fuse the coarse to fine TSDF volumes using sparse attention in the so-called "top-down-bottom-up” approach. The experiments shows ablations and comparisons where the proposed approach performs well.

**Summary Of The Review:**

My biggest concern is that the experiments and inference are not as well communicated. Despite this I am rating the paper above the acceptance threshold hoping that the authors will answer the questions/concerns.

---

> ### Author Response · Authors · 2022-11-18
> **Author Responses to Reviewer N9G6**
>
> Thanks for the valuable feedback and constructive comments. In the following, we address the reviewer's comments point by point.
>
> 1. ####  Relationship to previous work.
>
>     Thanks for the constructive advice.
>     We had some descriptions of the relations to previous work.
>     In the introduction section, we wrote "In this paper, we follow this direct 3D reconstruction formulation", which means we follow approaches like Atlas and NeuralRecon.
>     In the method section, we wrote "following Murez et al.(2020), we back project ...".
>
>     We would like to add more descriptions about the relation to previous methods in the revision.
>     Please see the modified introduction: This way of reconstruction is end-to-end trainable, and is demonstrated to output accurate, coherent, and complete meshes.
>     In this paper, we follow this volume-based 3D reconstruction path and directly regress the TSDF volume.
>     Please see the modified related work: In this work, we follow this volume-based architecture to directly regress the 3D shape from a TSDF representation in a coarse-to-fine manner, but enhance it with a proposed 3D transformer structure.
>
>
> 2. ####  Discussions on what does not work when directly integrating the transformers.
>
>     As stated in the paper, the biggest problem of directly integrating the transformers is the high computation complexity, which consumes too much memory such that it cannot be realized in regular GPUs, even in NVIDIA A100 with a memory of 80G.
>     The computation complexity of a direct 3D transformer is $\mathbb{O}_\text{3D-Attn} = N_X \times N_Y \times N_Z \times N_X \times N_Y \times N_Z \times \mathbb{O}(ij)$, which is quite large. Therefore, previous approaches mainly adopt only 2D transformers to process 2D features.
>     We would like to add more discussions on this in the revision. Please see Section 3.3 in the revised manuscript.
>
> 3. ####  Loosely written phrases.
>
>     Thanks for these constructive suggestions. We have changed to "SLAM [1] or SfM [2] systems" in the modified manuscript.
>     In the experiments section, the citation of NeuralRecon was written in Table 1 and Table 2. The missing in some places is for saving space. We have added them back in the modified manuscript, such as the caption of Table 1.
>     In addition, more details are provided in the experiments of the modified manuscript.
>     More suggestions are welcome!
>
> 4. ####  Sequence length used for the experiments at a time.
>
>     Sorry for the unclear statement. We have added more details to the modified manuscript in section 4.1 and appendix A.1:
>     Actually, any number for the sequence length is ok for our framework, although more images lead to a better reconstruction of a scene.
>     In the training, the number for the sequence length is set to 20, i.e., twenty images are input to the framework in a single iteration.
>
> 5. ####  More details about inference.
>
>     Sorry for the unclear statement. We have added more details about inference to the modified manuscript in section 4.1 and appendix A.1:
>     The inference is performed in the same way as the training.
>     For a given sequence of images, first a view selection is performed to select images with translation greater than 0.1m and rotation greater than 15 degrees.
>     Then a random selection is adopted from the remaining images if the number exceeds the upper limit. These images are then fed to the 2D backbone for feature extraction, after which the features are fused to a 3D volume and fed to the 3D part to produce the TSDF volume prediction.
>     The final mesh is extracted by marching cubes from the TSDF volume.
>     This process is the same as previous methods like Atlas, TransformerFusion or VORTX.
>     The difference between the inference and the training is the number limit of images: 20 for training and 100 for inference.
>
> 6. ####  Texture-less rendering of the GT meshes.
>
>     Thanks for this good suggestion. Due to the limited space of the main body, it is not enough to show both textured and texture-less GTs.
>     Following the suggestion of the reviewer, we have added the texture-less GTs to Figure~6 in the appendix in the modified manuscript.
>
> 7. ####  Limitation section.
>
>     Thanks for the advice. Actually, we mentioned the limitations of our methods in section A.4 of the appendix due to the limited space of the main body:
>     Due to the volume representation, our framework is limited by the trade-off between the resolution of the volume and the memory consumption.
>     A smaller voxel size would cost much more memory.
>     The voxel size is set to $4$ cm, such that the geometry details less than $4$ cm are hard to be recovered.
>
>
> [1] Campos C, Elvira R, Rodríguez J J G, et al. Orb-slam3: An accurate open-source library for visual, visual–inertial, and multimap slam. IEEE Transactions on Robotics, 2021.
>
> [2] Schonberger J L, Frahm J M. Structure-from-motion revisited. CVPR. 2016.

---

> > ### Comment · Reviewer_N9G6 · 2022-12-13
> > **Updated score**
> >
> > Thank you for the answers. Overall I understand the work much better now. The additional information provided on training and inference were of particular importance to me, as it was assumed to be obvious previously. I appreciate the authors' changes in the main paper and the appendix. Overall I think the paper is a small step forward in multiview SDF reconstruction architecture.

---

### Official Review · Reviewer_9DEs · 2022-10-31

**Confidence:** 3
**Correctness:** 4
**Technical Novelty And Significance:** 2
**Empirical Novelty And Significance:** 2
**Recommendation:** 6

**Clarity, Quality, Novelty And Reproducibility:**

This paper is generally well-written, and I believe it's not difficult to implement.

Questions:
Is this method heavily relied on the accuracy of pose estimation? Is it robust to reconstruct the scene when the poses of several frames are miscalculated?

**Strength And Weaknesses:**

Strength:
+ The approach is reasonable.
+ Paper is well organized and easy to understand.
+ Ablation study is extensive. And the results validate the proposed modules.


Weaknesses:
- The contributions are slightly incremental. Most approaches are introduced from existing works. The sparse window module is a standard technique in volume-based methods. Global attention and context encoding are commonly-used in 2D tasks.
- The effect of dilation-attention is doubtful. Intuitively, the dilation and the downsampling may lose more geometry information, while the paper explains that the geometry structure benefits from dilation.

**Summary Of The Paper:**

This paper proposes an SDF transformer network to improve monocular scene reconstruction. Firstly, the 3D transformer is introduced to aggregate the 3D features at different levels in a coarse-to-fine pattern. Secondly, a sparse window multi-head attention module is adopted to save computation costs. Thirdly, the dilate-attention structure, the global attention module, and the global context encoding module are designed to further improve the performance. The results of the ScanNet dataset demonstrate the effectiveness of the proposed method.

**Summary Of The Review:**

The paper aims to improve monocular scene reconstruction with 3D SDF transformers. A series of techniques are adopted to enhance the features. Experimental results validate the effectiveness of all the proposed modules. Considering the high quality of paper and incremental contributions, my current recommendation is marginally above acceptance.

---

> ### Author Response · Authors · 2022-11-18
> **Author Responses to Reviewer 9DEs**
>
> Thanks for the valuable feedback and constructive comments. In the following, we address the reviewer's comments point by point.
>
> 1. #### Contribution.
>
>     Thanks for the constructive comment of our reviewer and we would like to highlight the novelty of our proposed work.
>     As accurately mentioned by Reviewer Zo2i, although the modules (1,2,4) "are not ideas hard to figure out individually, it is good to put them together to build the 3D reconstruction system, and investigate their effects to the reconstruction quality".
>     Also, module (3) "is something novel", with which we build the first top-down-bottom-up 3D transformer network.
>     Notably, as stated in the introduction. this is the first paper employing the 3D transformer for 3D scene reconstruction from a TSDF representation, while previous approaches mainly stay at the 2D transformer.
>     In the following, we directly cite the reviewing comments of Reviewer N9G6, Zo2i and mQ2G, which acknowledge our novelty and contributions:
>
>     Reviewer N9G6: "Although the novelty may be sufficient, it is advisable to ..."
>
>     Reviewer Zo2i: "This work has good technique contributions"; "The approach is novel. Although 1), 2) and 4) are not ideas that are hard to figure out individually, it is good to put them together to build the 3D reconstruction system, and investigate their effects to the reconstruction quality. Moreover, I think idea 3) is something novel which might inspire or followed by other 3D sparse volume UNet style work ..."
>
>     Reviewer mQ2G: "the paper is is solid and makes a worthwhile contribution to the community"
>
>
> 2. #### The effect of dilation-attention.
>
>     Yes, the dilation operation alone may lose the geometry information, since it may add some wrong voxels into the sparse structure.
>     But here we equip the attention module next to the dilation operation, such that the voxels far from the surface would get low scores and do not contribute to the final shape.
>     This operation is to prevent the geometry shrinking in the down-sampling. Our sparse attention module can handle geometry dilation but has difficulty in handling geometry shrinking, because the lost voxels cannot be added back after being labeled as empty.
>     For example, the leg of the chair shown in Figure 4 cannot be recovered unless a dilate-attention module is added at the last.
>     We have added this explanation in Section 3.4 in the revised manuscript.
>
> 3. #### Robustness to the pose noise.
>
>     Our method is based on the given accurate camera poses, which is the same as previous state-of-the-art methods like Atlas, NeuralRecon, and TransformerFusion, where the camera poses are obtained by the standard SfM or SLAM systems.
>     To inspect the robustness of our method to the pose errors, we add the Gaussian noise to the camera poses. A translation noise $[N_x, N_y, N_z]$ of $N=Gauss(0, \sigma_T)$ is added to the translation of the pose, while a rotation noise $[N_{roll}, N_{pitch}, N_{yaw}]$ of $N = Gauss(0, \sigma_R)$ is added to the three angles of the pose.
>     The metrics following NeuralRecon are reported in the below table. From the results, we can see our system can handle some translation errors but cannot handle the rotation errors well.
>     But if the poses of only some frames are miscalculated, e.g., 10\% of all frames, the performance decrease would be under control.
>
>     | Ratio  | $\sigma_T$ (cm)  | $\sigma_R$ (deg)  | Acc $\downarrow$   | Comp $\downarrow$   | Prec $\uparrow$   | Recall $\uparrow$   | F-score $\uparrow$   |
>     | ----------- | ----------- | ----------- | ----------- | ----------- | ----------- | ----------- |----------- |
>     | 0 | 0 | 0 | 0.049 | 0.068 | 0.754 | 0.664 | 0.705 |
>     | 100\% | 0.5 | 0 | 0.050 | 0.068 | 0.745 | 0.658 | 0.698 |
>     | 100\% | 1  | 0 | 0.055 | 0.073 | 0.708 | 0.626 | 0.663 |
>     | 100\% | 0  | 0.5 | 0.084 | 0.117 | 0.525 | 0.446 | 0.480 |
>     | 100\% | 0  | 1 | 0.109 | 0.185 | 0.406 | 0.314 | 0.351 |
>     | 100\% | 0.5 | 0.5 | 0.084 | 0.117 | 0.516 | 0.435 | 0.471 |
>     | 100\% | 1  | 1 | 0.114 | 0.187 | 0.380 | 0.296 | 0.330 |
>     | 10\% | 0.5 | 0.5 | 0.055 | 0.074 | 0.715 | 0.629 | 0.668 |
>     | 10\% | 1  | 1 | 0.064 | 0.087 | 0.662 | 0.577 | 0.616 |
>
>     Table 1. Experiments with pose noise following NeuralRecon metrics.

---

### Author Response · Authors · 2022-11-18
**Paper Revision**

We thank all reviewers for their valuable feedback and constructive comments. We have revised the paper as suggested by the reviewers, and summarize the major revisions as follows:

- More description of the relationship to previous works, in Section 1 and Section 2.

- More detailed description of the SDF transformer model, including the network structure and the dilate-attention module, in Section 3.3 and 3.4.

- More detailed description of the experiments and inference, in Section 4.1 and Appendix A.1.

- Texture-less rendering of the ground-truth meshes, in Appendix A.7.

- Experiments of larger window size, in Section 4.3.

- Experiments of noisy camera poses, in Appendix A.5.

- Experiments of smaller voxel size, in Appendix A.6.

- Loosely written phrases. Some citation problems. Some notation problems.

The other concerns have also been addressed individually in the Author Responses.

---

### Decision · Program_Chairs · 2023-01-20

**Decision:**

Accept: poster

**Justification For Why Not Higher Score:**

This paper, while obtaining impressive results based on the proposed 3D Transformer, does closely follow the overall outline of prior methods.

**Justification For Why Not Lower Score:**

The work demonstrates impressive results/improvements on a core 3D task, and the memory-efficient 3D transformer may also be of intrinsic interest.

**Metareview: Summary, Strengths And Weaknesses:**

The paper tackles the task of dense 3D reconstruction from monocular sequences, and introduces a transformer-based architecture for inference. While the overall approach follows prior methods, the key contributions relate (efficiently) incorporating 3D transformers for the multi-view aggregation. The reviewers are all positive about the contributions and empirical results, and while there were some clarity concerns expressed by two of the reviewers, these have been adequately addressed by the revisions.  Beyond the task studied here, the AC also feels that the proposed Transformer design could also be informative for other 3D prediction/reconstruction tasks.


**Note From Pc:**

if the above contains the word "oral" or "spotlight" please see: "oral" presentation means -> notable-top-5% and "spotlight" means -> notable-top-25%. As stated in our emails, we are disassociating presentation type from AC recommendations